# High-Strength, Self-Healing Copolymers of Acrylamide and Acrylic Acid with Co(II), Ni(II), and Cu(II) Complexes of 4′-Phenyl-2,2′:6′,2″-terpyridine: Preparation, Structure, Properties, and Autonomous and pH-Triggered Healing

**DOI:** 10.3390/polym16223127

**Published:** 2024-11-09

**Authors:** Evgeny S. Sorin, Rose K. Baimuratova, Mikhail V. Zhidkov, Maria L. Bubnova, Evgeniya O. Perepelitsina, Ainur F. Abukaev, Denis V. Anokhin, Dmitry A. Ivanov, Gulzhian I. Dzhardimalieva

**Affiliations:** 1Federal Research Centre of Problems of Chemical Physics and Medicinal Chemistry RAS, 142432 Chernogolovka, Russia; rozab@icp.ac.ru (R.K.B.); zhidkov@icp.ac.ru (M.V.Z.); bml@icp.ac.ru (M.L.B.); jane@icp.ac.ru (E.O.P.); ainurabukaev@gmail.com (A.F.A.); anokhin@icp.ac.ru (D.V.A.); dimitri.ivanov.2014@gmail.com (D.A.I.); 2Department of Chemistry, Lomonosov Moscow State University, 119234 Moscow, Russia; 3Institut de Sciences des Matériaux de Mulhouse (CNRS UMR 7361), 68057 Mulhouse, France; 4Moscow Aviation Institute, National Research University, 125993 Moscow, Russia

**Keywords:** self-healing, metal acrylates, metal-containing monomers, supramolecular polymers, terpyridine-containing polymers, metallopolymers

## Abstract

The utilization of self-healing polymers is a promising way of solving problems associated with the wear and tear of polymer products, such as those caused by mechanical stress or environmental factors. In this study, a series of novel self-healing, high-strength copolymers of acrylamide, acrylic acid, and novel acrylic complexes of 4′-phenyl-2,2′:6′,2″-terpyridine [Co(II), Ni(II), and Cu(II)] was prepared. A systematic study of the composition and properties of the obtained polymers was carried out using a variety of physicochemical techniques (elemental analysis, gel permeation chromatography (GPC), attenuated total reflectance Fourier transform infrared spectroscopy (ATR/FT-IR), ultraviolet-visible spectroscopy (UV-vis), small-angle X-ray scattering (SAXS), wide-angle X-ray scattering (WAXS), differential scanning calorimetry (DSC), thermogravimetric analysis (TGA), dynamic mechanical analysis (DMA), confocal laser scanning microscopy (CLSM), and tensile testing). All metallopolymer samples exhibit autonomous intrinsic healing along with maintaining high tensile strength values (for some samples, the initial tensile strength exceeded 100 MPa). The best values of healing efficiency are possessed by metallopolymers with a nickel complex (up to 83%), which is most likely due to the highest lability of the metal–heteroatom coordination bonds. The example of this system shows the ability to re-heal with negligible deterioration of the mechanical properties. The possibility of tuning the mechanical properties of self-healing films through the use of different metal ions has been demonstrated.

## 1. Introduction

Self-healing materials are capable of restoring their original characteristics after damage [1]. To date, healing has been demonstrated for a variety of materials such as ceramics [2], metals [3], concrete [4], polymer systems [5], and others.

In terms of the need for initiation, the healing processes occurring in such materials can be divided into autonomous and non-autonomous, requiring different external stimuli such as microwave irradiation, UV-visible radiation, heating, magnetic field, pH, or the addition of initiating agents [6]. The classification by the nature of the process according to which recovery can be extrinsic and intrinsic is more important in terms of the self-healing materials’ design in general [5]. External healing involves the introduction of various healing agents into a material, allowing, for example, the creation of self-healing metals and ceramics [7,8]. Intrinsic healing, on the other hand, is due to the presence of various reversible interactions in the structure such as hydrogen bonds [9], π-π stacking interactions [10], metal-ligand coordination bonds [11], etc.

It is regrettable that there are instances when it is not feasible to render a material intrinsically healable. Consequently, it becomes necessary to incorporate healing agents into the finished product. In turn, this places a limit on the number of possible healing cycles, minimizing it. Once active agents are released from microcapsules or vascular networks [12], the material loses its ability to repair itself. Therefore, the design of materials that allow healing through the reversible rupture and the formation of dynamic bonds is a possible solution to the challenges of extending product life.

In this respect, polymers are the most widely used materials for the self-healing systems’ design due to the possibility of introducing various reversible interactions into their structure (e.g., π-π stacking and coordination interactions [10,11]). At the same time, the very presence of hydrogen bonds and van der Waals forces [13], which have low energy separately, leads to strong intermolecular interactions in polymer chains. Because of this, many polymeric materials themselves have the potential to self-heal and are able to repair in this way by an intrinsically healing mechanism. Thus, some of the first studies in this field focused on the crack-healing effect in polyvinyl acetate, carried out under the direction of V.A. Kargin [14].

Metal–ligand (M–L) coordination interactions are often used to increase the number of reversible dynamic bonds in a polymer, as a wide range of both metal ions and ligands allows tuning the bond strength and, thus, the physical and mechanical characteristics of the material as a whole [15]. It should be noted that the development of self-healing polymers was inspired by a variety of biological systems in many early studies [16]. The mechanism of self-healing in mussels was shown to be based on reversible coordination interactions between iron ions and catechol ligands [17], which prompted the development of a class of self-healing metallopolymers with reversible M–L interactions. In addition, the presence of coordination interactions often leads to the appearance of sensitivity of the polymer to various external influences, which makes it possible to create various stimulus-sensitive materials [18].

To date, a number of self-healing metallopolymers have been synthesized, the healing of which was attributed to the presence of coordination bonding of metal ions with various organic ligands (for example, with carboxylic acids [19], with phenolic ligands [20], etc.), but the most favored are interactions with pyridine [21,22] and its derivatives, in particular, with terpyridines [23,24,25,26], due to their high binding capacity.

Despite the wide variety of metallopolymers containing terpyridine moieties with promising potential applications (e.g., [27,28]), there are currently very few examples of self-healing polymers with intrinsic healing abilities and high strength properties (some of the highest values reported in [24] reach up to 50 MPa). Probably, this is due to the method of producing such self-healing metallopolymers based on the impregnation of pre-made polymer hydrogels (containing various chelating ligands) with salts of different transition metals [24,28,29,30]. On the one hand, strong coordination interactions and increased cross-linking decrease the solubility of hydrogels, often making it difficult to obtain a high-quality film. On the other hand, these interactions are generally not sufficient to significantly improve the strength properties of the system.

Hence, we have previously proposed a one-step method for the preparation of self-healing copolymers of acrylamide (AAm), acrylic acid (AAc), and novel metal acrylate complexes with 4′-phenyl-2,2′:6′,2″-terpyridine (PhTpy) [31,32]. The key feature of these systems is the incorporation of the metal ion directly into the polymer chain. Acrylamide and acrylic acid were chosen as model copolymers because the self-healing of their hydrogels is a frequent example of healing effects in polymers [33]. The interaction of carboxyl and amide groups with metal ions also leads to the formation of self-healing metallopolymers [34].

In the previous work, the fundamental possibility of obtaining high-strength self-healing polymer films has been demonstrated [32]. The main objectives of this study were to extend this class of self-healing copolymers using novel acrylate complexes of 4′-phenyl-2,2′:6′,2″-terpyridine [Co(II), Ni(II), and Cu(II)] and to evaluate the effect of the inclusion of metal chelate monomers on the structure and self-healing properties of acrylamide and acrylic acid copolymers.

Therefore, this paper provides an extensive comparative analysis of the physicochemical properties of copolymers containing phenylterpyridine complexes with acrylates of various transition metals and the evaluation of the effect of metal ion on the structure and self-healing properties of the obtained polymers. Furthermore, the preparation of model systems by copolymerization of AAm and AAc without a metal complex as well as with cobalt acrylate (CoAcr_2_) allowed to quantify the influence of the inclusion of both a metal ion and a phenylterpyridine ligand, separately, on the recovery processes. The results demonstrate that the highest healing efficiency values are observed when the metal complex with phenyltepyridine is included in the chain compared to the model systems. The tensile strength values of some samples exceed 100 MPa, and autonomous intrinsic healing is observed for all metallopolymer samples—the crack heals in three days in air at room temperature in a humid atmosphere. The possibility of secondary healing of the specimen is also demonstrated. The best efficiency of tensile strength recovery during pH-triggered healing was observed in a series of samples containing nickel complexes, which is most likely due to them having the highest lability of metal–heteroatom coordination bonds.

## 2. Materials and Methods

### 2.1. Starting Materials

Acrylamide (AAm, >99%), cobalt (II) chloride hexahydrate (CoCl_2_∙6H_2_O, 98%), nickel (II) chloride hexahydrate (NiCl_2_∙6H_2_O, 98%), copper (II) carbonate hydroxide ((CuOH)_2_CO_3_, 98%), and potassium persulfate (K_2_S_2_O_8_, ≥99.0%) were purchased from Sigma-Aldrich (Moscow, Russia) were used without additional purification. Acrylic acid (AAc stabilized with hydroquinone monomethyl ether for synthesis, >98%) purchased from Sigma-Aldrich (Moscow, Russia) was used after additional purification by vacuum distillation. The preparation and the physical and chemical properties of metal-containing monomers based on AAc and PhTpy were described earlier [35,36,37]. The general scheme of monomer production is presented in Figure 1.

### 2.2. Synthesis of Cobalt Acrylate

The cobalt acrylate required for the preparation of phenylterpyridine-free metallopolymers was prepared according to the following procedure.

A solution of 1.40 g Na_2_CO_3_ (13.2 mmol) in 20 mL of water was added to a solution of 1.57 g CoCl_2_∙6H_2_O (6.6 mmol) in 20 mL of water under stirring (temperature of both solutions 40 °C), after which the reaction flask was tightly corked to prevent carbon dioxide from escaping. The cobalt carbonate was separated by filtration using a Buechner funnel and rinsed until the wash water was neutral.

To a suspension of freshly prepared cobalt carbonate in water at room temperature, 0.9 mL (0.95 g, 13.2 mmol) of acrylic acid was added with stirring, and the mixture was left to stir for one hour under inert atmosphere. The resulting pink-colored solution was filtered under an inert atmosphere. After that, the target product was isolated by distilling the solvent under vacuum (1.03 g, 5.1 mmol, yield: 77%).

The results of the elemental analysis of the compound obtained are close to the theoretical values (found content, %/theoretical content, %): C—34.76/35.80, H—2.98/3.63).

### 2.3. Synthesis of Polymers and Film Formation

Polymers were prepared by free-radical polymerization of an aqueous precursor solution containing certain amounts of acrylamide (AAm), acrylic acid (AAc), and acrylate complexes of phenylterpyridine (MAcr_2_PhTpy; M = Co(II), Ni(II), and Cu(II)) with a polymerization temperature of 60 °C and the initiator being K_2_S_2_O_8_.

Since the room temperature polymerization of 50% aqueous solutions of monomers occurred in the frontal mode with high heat generation [32], 20% aqueous solutions were used to obtain copolymers in this work. Copolymer systems containing complexes of phenylterpyridine with acrylates of cobalt (Co-Copolymers series), nickel (Ni-Copolymers series), and copper (Cu-Copolymers series) were prepared. To investigate the effect of the ratio of acrylamide to acrylic acid on the final properties of the polymers, each series consisted of three monomer blends in which acrylamide was in excess, in equimolar ratio, and in deficiency with respect to acrylic acid (Table 1).

Copolymers of acrylamide and acrylic acid with cobalt acrylate (CoAcr_2_) were also prepared to analyze the phenylterpyridine’s role in self-healing processes (CoA-Copolymers series). In addition, for a comprehensive study of the influence of the metal ion inclusion directly into the polymer chain (in the form of acrylate or in the form of metal complex) on the final structure and properties of polymers, this paper also includes studies of previously obtained copolymers of acrylamide and acrylic acid, not containing phenylterpyridine complex (series of Copolymers) [32]. For the sake of convenience, these copolymers will, henceforth, be referred to as «initial». The compositions of precursor solutions for these model systems are presented in Table 2. It should be noted that the required amount of cobalt acrylate was calculated for equimolarity reasons with respect to its complex with phenylterpyridine for correct comparison of the self-healing properties of the obtained metallopolymers.

The films were obtained according to the methodology described in the previous work [32]. The films were formed from an aqueous polymer solution in special open glass molds by air drying at room temperature. The drying process was carried out until the film mass reached a constant value. Once the film had completely detached from the mold, it was inverted to ensure maximum drying on both sides.

### 2.4. Analytical Methods

#### 2.4.1. Analysis of Structure and Thermal Properties

C, N, and H content was studied on a Vario Micro cube elemental analyzer (Elementar GmbH, Hanau, Germany), and the content of cobalt, nickel, and copper was studied on an AAS-3 atomic absorption spectrometer (Zeiss, Jena, Germany).

IR analysis was performed on a Bruker ALPHA Fourier-IR spectrometer (Bruker Optik GmbH, Ettlingen, Germany) equipped with a single reflection diamond prism; the penetration depth for a medium with a sufficiently deep refractive index (2.43) at 1000 cm^−1^ is 1.66 microns. UV spectrometry was carried out on a SPECS-SSP-705-1 spectrometer (JSC Spectroscopic Systems, Moscow, Russia).

Analysis of the molecular weight distribution was carried out on a «Waters» liquid chromatograph equipped with a «Waters2414» differential refractometric detector and a «PDA 996» diode array spectrophotometric detector (Waters Corporation, Milford, MA, USA). A PLgel 5 μm MIXED-C column was used. The eluent used was N-methylpyrrolidone (NMP) + LiCl (1.0 g LiCl/0.5 L NMP), and the elution rate was 1 mL/min; T_col_ = 70 °C, and T_ref_ = 50 °C. Standard polystyrene samples with MM from 580 to 3.7106 Da were used to construct the calibration curve. The obtained chromatograms were processed using the «Empower» software (Empower Pro 2002 Waters Corporation). The polymer samples were dissolved in NMP + LiCl (concentration 20 ÷ 40 mg/mL) and the solution was filtered through a 0.2 µm PTFE filter Anatop25 (“Whatman”). The polymer was dissolved at room temperature for 24 h, followed by a further 6 h at 55 °C. The polymer solution (ash part) was then carefully separated from the undissolved part of the polymer (precipitate/gel part) using a syringe. The polymer solution was further filtered, after which the filtrate was used for chromatography.

Small and wide-angle X-ray diffraction analysis (SAXS/WAXS) was carried out using a Xenocs Xeuss 3.0 diffractometer (Cu Kα, wavelength λ = 1.5418 Å) (Xenocs Inc, Grenoble, France). Two-dimensional diffraction patterns were captured using a 2D Dectris EIGER2 S 1M detector. The distance between the samples and detector was set at 70 mm (q-range 0.36–3.7 Å^−1^) and 1700 mm (q-range 0.004–0.21 Å^−1^) for WAXS and SAXS, respectively. To calibrate the modulus of the scattering vector, lanthanum hexaboride and silver behenate reflections were used. Data reduction was performed using a custom software environment.

Surface topology films were observed with a confocal laser scanning microscope (Optelics Hybrid, Lasertec Corp., Tokyo, Japan) equipped with 5×, 10×, and 20× objectives.

Differential scanning calorimetry (DSC) and thermogravimetric analysis (TGA) curves were recorded on a XiangYi Instrument DSC-200 differential scanning calorimeter (XiangYi Instrument Co., Ltd., Xiangtan, Hunan, China) and a METTLER TGA/SDTA851e thermogravimetric analyzer (Mettler Toledo, Greifensee, Switzerland). The samples were heated in a nitrogen atmosphere at a heating rate of 10 K∙min^−1^.

The dynamic mechanical properties of the specimens were measured using a dynamic mechanical analysis instrument DMA 242 C (Netzsch-Gerätebau GmbH, Selb, Germany) in tensile mode with continuous temperature scanning from 10 °C to 190 °C at a rate of 2 K∙min^−1^ in helium atmosphere. A sinusoidal oscillating force was applied to the specimens to develop a maximum strain amplitude of 30 μm, at a fixed loading frequency of 1 Hz.

#### 2.4.2. Physical and Mechanical Tests and Assessment of Self-Healing Efficiency

Physical and mechanical tests were carried out on a universal tensile machine Zwick/Roel TC-FR010 (ZwickRoell GmbH & Co. KG, Ulm, Germany) at room temperature at a tensile speed of 1 mm/min. For the study, two-sided blade-shaped samples of length L = 75 mm, tear-off width d = 4 mm, and an average thickness of 1 mm were cut from tear-off width (Russian GOST 270-75 type III) films. All series of samples were presented from more than five blades, and the discrepancy between the test results did not exceed 10%.

pH-triggered healing was carried out according to the following process. After the tensile test, the two parts of the ruptured sample were brought into contact and the rupture site was moistened with 0.1 M HCl to accelerate the healing process.

Healing efficiency was evaluated by comparing the tensile strength and elastic moduli of the healed specimen with the original specimen, according to Equations (1) and (2):(1)$$Z_{\sigma }=\frac{\sigma_{h}}{\sigma_{0}}\times 100\%,$$
(2)$$Z_{E}=\frac{E_{h}}{E_{0}}\times 100\%,$$
where σ_h_ and E_h_ are the tensile strength and modulus of elasticity of the healed sample and σ_0_ and E_0_ are the tensile strength and modulus of elasticity of the original sample.

Polymer films were placed in a desiccator with a small amount of water without contact with solvent to demonstrate autonomous internal healing. The samples were incubated at room temperature in a humid atmosphere for three days and then examined using optical and laser scanning microscopes.

## 3. Results and Discussion

### 3.1. Structure, Composition, and Thermal Properties of Copolymers

#### 3.1.1. Elemental Analysis and Gel Permeation Chromatography

The results of elemental analysis and gel permeation chromatography are presented in Table 3.

The results of the copolymer composition calculation indicate that as the acrylamide concentration within the system rises, the residual moisture content within the films also increases. This may be caused by a more pronounced interaction of acrylamide groups with solvent molecules due to the formation of hydrogen bonds [38]. In general, the films contain 10–15% residual solvent after drying. It is possible that this feature of these systems may have an adverse effect on the stability of their physical and mechanical characteristics. Conversely, it may also encourage the segmental mobility of polymer chains and facilitate ion exchange processes at the contact points of contacting surfaces [39], thus potentially facilitating the healing process itself.

Regarding the investigation of the molecular weight characteristics of the obtained copolymers, account should be taken of the fact that the experimental data were obtained by gel permeation chromatography of the ash part. In this respect, the data presented in Table 3 do not represent the true values of the molecular-mass characteristics of these copolymers. However, these data may, on the one hand, be used as evidence of the high molecular weight product formation, and on the other hand, be employed to provide a relative assessment of the impact of the monomer mixture composition on the molecular weight characteristics of the obtained copolymers.

Thus, M_w_ and M_n_ increase with increasing acrylic acid content for copolymer series with copper and nickel complexes. At the same time, Juárez et al. demonstrated that when the AAm:AAc ratio is varied from 25:75 to 75:25, the M_w_ values do not change significantly [40]. Probably, the decrease in the obtained values of M_w_ and M_n_ with increasing acrylamide content for Cu-Copolymer 3 and Cu-Copolymer 2 copolymers, as well as for Ni-Copolymer 3 and Ni-Copolymer 2, is due to the analysis of their ash content. Overall, it was not possible to isolate the ash part for all copolymer series at monomer ratio AAm:AAc = 85:15. Moreover, the low solubility of the model copolymer hydrogel (with cobalt acrylate/without terpyridine complex) at excess acrylamide makes it impossible to produce a film. The possibility of obtaining polymer films with an excess of acrylamide upon addition of the metal complex is most likely due to the appearance of additional free volume due to the inclusion of sterically hindered phenylterpyridine ligand, which may also have a positive impact on self-healing processes.

Alternatively, the molecular weight characteristics’ decrease may be attributed to the different strength of coordination of the monomer with primary radicals, leading to their deactivation and to a decrease in the initiation efficiency. Previously, Pomogailo D.A. et al. [41] demonstrated that metal-containing monomers can participate in chain breakage reactions due to coordination with reaction centers and their subsequent deactivation. As a result, it contributed to the decrease of polymer molecular weight and degree of polymerization.

The values of M_w_ and M_n_ differ most strongly for Co-Copolymer 2 and take the minimum values among all obtained. This is probably due to the highest reactivity of the cobalt complex during copolymerization with AAm:AAc, which, in turn, leads to an increase in the degree of crosslinking (as well as to cyclization processes, the products of which were observed during the study of polymers by infrared spectroscopy (Section 3.1.2)) and a decrease in the amount of the ash part of the copolymer due to slower and incomplete extraction. As mentioned in the previous work, the polymerization reaction of 50% aqueous solutions of monomers upon the addition of cobalt complex and initiator proceeded at room temperature in the frontal mode. Among the 20% aqueous solutions used in this study, the reaction solution gelation visually started also almost immediately at room temperature in the case of copolymerization with cobalt complex. This feature is associated with the electronic effect of the metal, depending on the value of its electronegativity, on the unsaturated bond reactivity, as well as with the change in the degree of conjugation of the unsaturated bond with the carboxylate ion, depending on the nature of the metal [42,43]. This leads to an increase in the rate of polymerization of metal acrylates in the following series: Cu^2+^ < Ni^2+^ < Co^2+^.

From another perspective, the most significant decrease in the molecular weight of polymers with cobalt complexes may be due to the monomer’s participation in reducing the activation energy of initiator decay into free radicals or its involvement in the initiation of radical polymerization, as demonstrated for spatially hindered redox-active cobalt complexes [44,45]. The polydispersity index decrease (see Table 3) also indirectly suggests the ability of the Co(II) complex to catalyze the chain transfer reaction on the monomer.

#### 3.1.2. IR- and UV-Vis-Spectrometry

The IR spectra of the polymer films were obtained in the ATR mode. The spectra of copolymers containing cobalt acrylate complex with phenyterpyridine (Co-Copolymers) are presented in Figure 2. Spectra for the model systems are shown in Figure 3a («initial» Copolymers) and Figure 3b (CoA-Copolymers) to analyze the influence of the metal ion inclusion in the polymer chain on the final structure of the copolymers. The spectra of copolymers with nickel (Ni-Copolymers) and copper (Cu-Copolymers) complexes are presented in the Appendix A.

Broad absorption bands in the 2900–3600 cm^−1^ region are observed for all the samples, which is caused by the presence of intra- and intermolecular hydrogen bonds. The bands around 3340 and 3200 cm^−1^ are related to the asymmetric and symmetric valence vibrations of NH_2_ and are observed for all series of samples with excess acrylamide and with equimolar AAm:AAc ratio (all Copolymer 2 and Copolymer 3 series) [46], except for copolymer samples with cobalt acrylate, where the 3340 cm^−1^ band is not observed. It is important to note the shift of the 3344 cm^−1^ band in the «initial» equimolar copolymer (Figure 3a, Copolymer 2) to 3338 cm^−1^ for copolymers with cobalt and nickel complexes (Figure 2 and Appendix A, respectively), which may indicate the coordination of metal ions with NH_2_ groups [47]; for copolymers with copper complex (Appendix A), this shift is hardly observed at all. Also in this range, peaks in the region of 2960, 2930, and 2850 cm^−1^ are observed for all samples, referring to the valence asymmetric and symmetric vibrations of the aliphatic CH_2_ groups of the main chain of the polymer.

The peak in the region of 1698 cm^−1^ observed in all spectra of the copolymers with equimolar ratio and with excess acrylic acid (all Copolymer 2 and Copolymer 3 series) belongs to the valence vibrations of C=O carboxyl groups. The presence of intense signals (for all Copolymer 1 and Copolymer 2 series) in the region of 1651 and 1602 cm^−1^ refers to the vibrations of amide I (ν C=O + ν C-N) and amide II (ν CN + δ N-H), respectively [46,48]. For this region, it is interesting to note the shift of the C=O vibrational peak of the amide group from 1651 cm^−1^ in the «initial» copolymers (Figure 3a) to 1649 cm^−1^ for copolymers containing a metal center in the chain (Figure 2, Figure 3b, Appendix A). A similar shift for the C=O vibrations of the carboxyl group from 1698 cm^−1^ to 1693–1695 cm^−1^ can also be observed. Thus, it can be assumed that metal ions can also coordinate on the C=O groups of amide and carboxyl groups. The observed splitting of these peaks implies the presence of both coordinated and uncoordinated groups.

The ~1450 and ~1415 cm^−1^ absorption bands observed in the spectra for all series of samples refer to the strain vibrations of the aliphatic groups of the main chain free and bound to the carboxyl group, respectively. The appearance of bands at ~1205–1210 cm^−1^ and 1150–1160 cm^−1^ for all copolymers may be due to the interaction of in-plane OH bond bending and C-O stretching vibrations in neighboring carboxyl groups [46,49]. Differences in these regions are observed for copolymers with cobalt acrylate (CoA-Copolymer series, Figure 3b) and with its phenylterpyridine complex (Co-Copolymer series, Figure 2), which show bands at ~1260 cm^−1^ as well as 1010–1020 and 1080–1090 cm^−1^. The bands at 1010–1020 and 1080–1090 cm^−1^ are most likely due to C-O-C vibrations, indicating the formation of six-membered anhydride acid ring structures [50]. The peak at 1260 cm^−1^ could be attributed to imide ring formation [48].

In the frequency range below 1000 cm^−1^, a band of skeletal vibrations of the C-C bond in the main polymer chain is observed at 799 cm^−1^. The ~510 cm^−1^ absorption band, previously attributed to possible M-O vibrations [32], is also observed in the spectra of «initial» copolymers. Absorption bands in the region of around 640 cm^−1^ are also typical for all samples. Probably, these peaks are caused by deformation out-of-plane vibrations of -OH groups; therefore, this region is not of practical interest for analysis in this study.

The UV-vis spectra of the polymer films are presented in Figure 4.

For copolymers containing phenylterpyridine complexes, the absorption maxima are similar to those of the corresponding acrylate phenylterpyridine monomers, indicating that the coordination of the metal ion with the phenylterpyridine ligand is maintained during polymerization. This signal is lower for copolymers with a nickel complex, which may be associated with more labile metal–ligand bonds, according to the data of XRD analysis of complexes [36,37]. The model systems exhibit no analogous charge transfer signals in their absorption spectra (Figure 4d), which further substantiates the attribution of these peaks to M–N bonds within the phenylterpyridine complex.

#### 3.1.3. X-Ray Studies

The SAXS curves (Appendix A) exhibit no characteristic peaks indicating the absence of aggregation of metals and ordering of the polymers at the supramolecular level. On WAXS curves (Figure 5 and Figure 6) of all samples, one can see two broad, overlapped peaks at q_1_ = 1.36 Å^−1^ (d_1_ = 4.6 Å) and q_2_ = 2.77 Å^−1^ (d_2_ = 2.77 Å). These peaks can be attributed to the characteristic distance between the acrylic and acrylamide side groups (d_1_) and between the main chains (d_2_) in amorphous polymers. The absence of other sharp peaks shows that metals or polymers do not have ordered structures, in contrast to our previous study [32]. In addition, the WAXS curves of Copolymer 2 and series Ni-Copolymers demonstrate a narrow reflection at q = 1.28 Å^−1^ (d = 4.9 Å) that can be attributed to (001) reflection of triclinic structure potassium persulfate [51].

#### 3.1.4. Thermal Properties of Polymers

The results of DSC, TGA, and DTG for copolymers containing cobalt complexes (Co-Copolymers series) as well as for model systems containing cobalt acrylate (CoA-Copolymers series) and for «initial» copolymers (Copolymers series) are shown in Figure 7, Figure 8, and Figure 9, respectively. Similar studies on the series of copolymers with nickel complexes (Ni-Copolymers series) and copper complexes (Cu-Copolymers series) are presented in the Appendix A, respectively.

The decomposition of all samples can be divided into four stages. There is a partial solvent loss in the first stage, which occurs up to 150 °C; the mass loss is insignificant (up to 5%). Endothermic peaks in the 60–70 °C range can be seen in the DSC curves, whereas little or no endothermic effect is observed at excess acrylamide (AAm/AAc = 85/15) for the copolymer samples with phenylterpyridine complexes (Figure 7 and Appendix A). This is likely due to the strong intermolecular interaction of acrylamide polymers with solvent molecules, which was mentioned when elemental analysis results were discussed.

The second and third stages take place on average in the ranges of 150–230 °C and 230–320 °C. The second stage is more pronounced for copolymer series with equimolar ratio and with excess acrylamide, based on the higher decomposition rates (maxima in the region of 200–230 °C on DTG curves), while the third stage is more marked for copolymer series in the case of excess acrylic acid, as can be observed from the peak in the region of 270–285 °C on DTG curves of these samples. In this context, it is highly probable that endothermic peak in the region of 200 °C for Copolymer 2 and Copolymer 3 is associated with the imidization process, which involves mass loss caused by NH_3_ release, while the endothermic peak in the region of 284 °C is attributed to the formation of acid anhydrides accompanied by the release of H_2_O [52]. The peak in the region of 236–238 °C in the second stage of decomposition of the metallopolymers with equimolar ratio and with excess acrylamide is also observed, but no additional changes are noted in the TGA curves. It would seem that this endothermic peak is also linked to the imidization processes of coordination-bound amide groups, in connection with which it is observed at higher temperatures (as is known, various metal ions are employed to shield acrylamide copolymers against thermal degradation [38]). The complicated temperature dependence of the thermal effect in the range of 200–240 °C for copolymers containing cobalt acrylate or cobalt complex may be due to the presence of imide structures within the initial polymer chain, as discussed in Section 3.1.2.

The fourth main decomposition stage is observed above 320 °C and is associated with the decarboxylation and decomposition of the main polymer chain. Attention should be drawn to the increase in the temperatures of the endothermic effect maxima and decomposition rate at the inclusion of a metal center in the copolymer, which may also indicate additional binding of carboxyl groups by metal ions. An interesting peculiarity is also the fact that the maximum decomposition rate is observed for all samples containing equimolar amounts of AAm and AAc. Since no maximum at 285 °C is observed for equimolar ratio copolymers, it can be assumed that the thermal stability increase due to hydrogen bonds leads to an increase in the decomposition temperature of acrylate units. The increased overall rate of mass loss is caused by the superposition of the two processes. An exothermic maximum is also observed in the range of 390–400 °C for «initial» copolymers and copolymers with cobalt acrylate and nickel complex at acrylic acid excess. This peak can be attributed to the decomposition of the previously formed AAc anhydride [52].

It is rather difficult to determine the glass transition temperature by DSC for the obtained copolymers; therefore, dynamic mechanical analysis was performed on copolymer samples with a cobalt complex (Co-Copolymers). The results are presented in Appendix A and Appendix A. The extrapolated onset temperature of the α-transition (defined as the point of intersection of two tangents to the elastic modulus curve—onset E′) was taken as the glass transition temperature, since the temperature of the inflection point on the elastic modulus curve in the glass transition region (inflection E′) may be less reliable due to the specificity of the samples and the shape of the curves obtained.

Typically, the glass transition temperature of acrylamide and acrylic acid copolymers ranges from ~97 °C, which is the Tg of pure polyacrylamide, to ~123 °C, which corresponds to the glass transition of polyacrylic acid [53,54,55]. Both Tg estimates reported in this paper for copolymers with cobalt complex (Co-Copolymers) demonstrate lower glass-transition temperatures compared to pure copolymers. The observed glass-transition temperature decrease can probably be caused both by the increase in the amount of residual solvent and the increase in the side groups volume contributed by phenylterpyridine. It leads to an increase in the free volume fraction due to a decrease in the packing density of macromolecules. In addition, this phenomenon may also be associated with the molecular weights decrease.

It is worth taking notice of the fact that Tg increases as the content of acrylamide in the copolymer increases, and not the other way around, as is observed for pure copolymers. This phenomenon may be explained both by a more pronounced interaction of acrylamide groups with metal ions and with solvent molecules and by the formation of imide ring structures. Both processes result in a decrease in the segmental mobility, and consequently, in the flexibility of the macromolecule, leading to the glass-transition temperature increase.

### 3.2. Self-Healing Properties of Metallopolymers

#### 3.2.1. Autonomous Intrinsic Self-Healing

To test the self-healing effect, a through incision of the polymer film was made so that the overall sample integrity was not disturbed. This approach allows bringing the two parts of the specimen into contact as tightly as possible, repeating the original position (which is practically impossible to realize when the specimen is completely torn into two parts, which will be discussed later). Thereafter, the specimens were allowed to air at ambient temperature in a moist environment. After three days, the cracks of all metallopolymer samples were tightened. As an example, Figure 10 demonstrates the self-healing of the equimolar copolymer with a copper complex (Cu-Copolymer 2). All specimens were examined both before and after the healing process using an optical microscope, and surface integrity after the healing was confirmed by confocal laser scanning microscope observation.

As can be seen from Figure 10, the crack in the specimen has completely healed after three days, leaving almost no visible trace in its place. Confocal imaging of this area also confirms complete surface recovery.

The results of a series of experiments designed to verify the autonomous self-healing effect are presented in Appendix A. It is noteworthy that autonomous intrinsic healing also occurs in CoA-Copolymers samples containing cobalt ions in the polymer chain without the phenylterpyridine ligand (Appendix A). At the same time, such an effect is practically not observed in «initial» copolymers (as it was shown in the previous work [32]), which confirms a significant contribution of coordination interactions to the healing processes. Weaker hydrogen bonds and π-π-stacking interactions are known to dissipate strain energy through reversible rupture and formation, while coordination bonds appear to provide efficient self-healing, by analogy, as shown for disulfide bonds [56].

#### 3.2.2. Mechanical Properties and pH-Triggered Healing Efficiency

One of the traditional methods for assessing the self-healing efficiency is to compare the values of the maximum strength and elasticity modulus of the healed sample (σ_h_, E_h_) with those of the original sample (σ_o_, E_0_), according to Equations (1) and (2). In this study, the healing process was accelerated by using 0.1M HCl solution, resulting in an increase in hydrogen bonding interactions due to the protonation of the side groups [57]. After the rupture tests, the two parts of the ruptured specimen were brought into contact and the rupture area was moistened with 0.1M HCl to accelerate the healing process.

Tensile tests were conducted on a minimum of five samples for each copolymer. The discrepancy between the experiments did not exceed 10%. The physical and mechanical characteristics of the obtained copolymers, as well as their healing efficiency, are presented in Table 4. The test results indicate that all polymer specimens exhibit high strength characteristics and a pH-triggered healing effect.

Following the acceleration of the healing process through the use of HCl, the specimen was transferred to a calcium chloride desiccator in order to facilitate a complete dehydration process. This appeared to reduce the moisture amount in the sample below the initial values, resulting in some polymer samples having a higher elasticity modulus after recovery. For such samples, the values of Z_E_ are not provided in the table. Accordingly, subsequent deliberations on the efficacy of self-healing will focus on the restoration of the strength values in lieu of the elastic modulus. It is important to note that, in order to assess the effectiveness of the healing process from the perspective of restoring the original structure, it is necessary to compare the tensile strengths (or maximum strengths) of the original and healed specimens in order to determine their resistance to critical stresses and the possibility of practical application. This is explained by the fact that the strength, in this case, is determined by the cross-linking of the specimen, while the elastic modulus is determined by the kinetic flexibility of the chains [58].

It is crucial to highlight that the healing efficiency of copolymers with cobalt complexes (Co-Copolymer 2, Co-Copolymer 3) is superior to that of the corresponding model copolymers comprising solely cobalt ions in the chain (CoA-Copolymer 2 and CoA-Copolymer 3). Furthermore, the «initial» model copolymers exhibit even lower healing efficiencies [32] than CoA-Copolymer 2 and CoA-Copolymer 3 metallopolymers (without the phenylterpyridine ligand). It can, thus, be concluded that the incorporation of transition metal ions into poly(AAm-co-AAc) chains can both enhance the efficiency of HCl-accelerated healing and, generally, achieve autonomous intrinsic healing of high-strength polymers (Appendix A). This phenomenon is most likely attributable to the presence of additional reversible coordination interactions between the cobalt ion and carboxyl and amide groups, as previously discussed.

The incorporation of phenylterpyridine metal complexes into polymer chains allows to enhance the efficiency of the healing process by up to twofold in comparison with metallopolymers comprising solely metal ions (for the comparison of Co-Copolymer 3 and CoA-Copolymer 3). This phenomenon may be attributed to a number of factors. The presence of phenylterpyridine in the polymer chains results in the interaction of π-orbitals (π-π-stacking), thereby augmenting the number of reversible interactions within the system. An important consequence of the inclusion of sterically hindered ligand into the polymer chains is also an increase in free volume, which has a beneficial impact on the healing process due to the enhanced flexibility of the macromolecule.

The results of the tensile tests indicate a classical balance, whereby the healing efficiency increases with a reduction in strength characteristics. However, it is notable that even the highest strength series of copper complex copolymers (Cu-Copolymers) demonstrate strength recovery of up to 30% with HCl-accelerated healing.

It is also noteworthy that the series of copolymers with the nickel complex (Ni-Copolymers) exhibits the highest healing efficiency, reaching up to 83%. This finding is consistent with the previously reported XRD results for monomers [36,37], which indicate that the nickel complex exhibits longer M–N bond lengths, which indicates their increased lability necessary for reversible processes.

For the series of these copolymers (Ni-Copolymers), tests were also carried out on the effectiveness of re-healing similarly accelerated by HCl, as one of the main advantages of intrinsic healing over extrinsic healing is the possibility of re-healing (Figure 11).

The tested specimens demonstrated the ability to re-heal with a recovery of up to 34% of the original strength after the second rupture (as well as a recovery of up to 59% of the strength value after the first rupture). Re-ruptured specimens occurred at the site of the original crack.

It is important to highlight that pH-triggered healing presents a number of distinctive characteristics in the context of metallopolymer systems, as noted in the previous work. The reaction of metal ions with hydrochloric acid can result in the release of these ions from the polymer chain. The consequence of such a reaction will be both an increase in the chains’ kinetic flexibility and the appearance of metal chlorides in the macromolecule free volume, the interaction of which with carboxyl and amide groups will then be facilitated by a decrease in the viscosity of the system. Actually, the situation outlined in the introduction, whereby self-healing polymers are obtained through impregnation with transition metal salts, is partially realized in the volume surrounding the crack. The presence of phenylterpyridine may also lead to additional rearrangements. All this leads to an acceleration of the self-healing processes, which generally took no more than 10 min with this approach (for example, see Figures 8–10 in [32]).

Furthermore, it is crucial to highlight the superior healing efficiency of copolymers with excess of acrylic acid in each series. Since the main chain of these copolymers contains the metal ion, the presence of such a feature may be indicative of a substantial role played by dissociation processes in the healing mechanism. In this case, the possibility of dissociation along the polymer chain due to the inclusion of metal is added to various classical dissociation processes of polyelectrolytes (which include copolymers of acrylic acid and acrylamide). It seems reasonable to propose that this phenomenon may also provide autonomous intrinsic healing.

It should be added that the method of assessing self-healing through tensile testing, while indicative, has the drawback of a partial loss of the original geometry of the sample in the rupture area. This phenomenon affects the underestimated results of recovering the original strength characteristics. For example, in this work, some samples recovered their surface almost without a trace during autonomous recovery in a humid atmosphere. However, it is virtually impossible to perform tensile tests in this case, as a constant tight fit (preferably identical to the initial film geometry) of the two completely separated specimens is required during healing. Nevertheless, some samples demonstrated good pH-triggered healing efficiency with record-breaking tensile strength values (over 100 MPa).

## 4. Conclusions

Novel self-healing, high-strength metallopolymers containing acrylic complexes of 4′-phenyl-2,2′:6′,2″-terpyridine [Co(II), Ni(II), Cu(II)] have been prepared. A detailed analysis of the obtained samples reveals the presence of a developed system of various reversible interactions in the copolymers, due to which, the effect of autonomous intrinsic healing can be achieved with the high strength of the samples. The highest healing efficiency is achieved in copolymers containing a phenylterpyridine complex, in comparison with the model «initial» copolymers and copolymers containing only metal ions in the chain. The best self-healing ability among metallopolymer series is possessed by copolymers with a nickel complex (Ni-Copolymers), which is most likely due to the highest lability of metal–heteroatom bonds. The ability to re-heal such systems has been demonstrated. The approach demonstrated in this study allows the production of high-strength, self-healing polymers with tunable physical and mechanical properties. An elaborate network of weak hydrogen and π-π-stacking interactions dissipates strain energy through reversible rupture and formation, while coordination bonds improve mechanical properties on par with providing efficient self-healing.

## Figures and Tables

**Figure 1 polymers-16-03127-f001:**
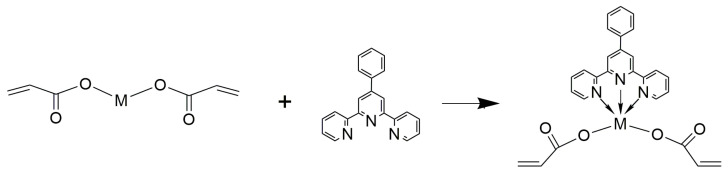
Scheme for the synthesis of acrylic phenylterpyridine monomers [M = Co(II), Ni(II), and Cu(II)].

**Figure 2 polymers-16-03127-f002:**
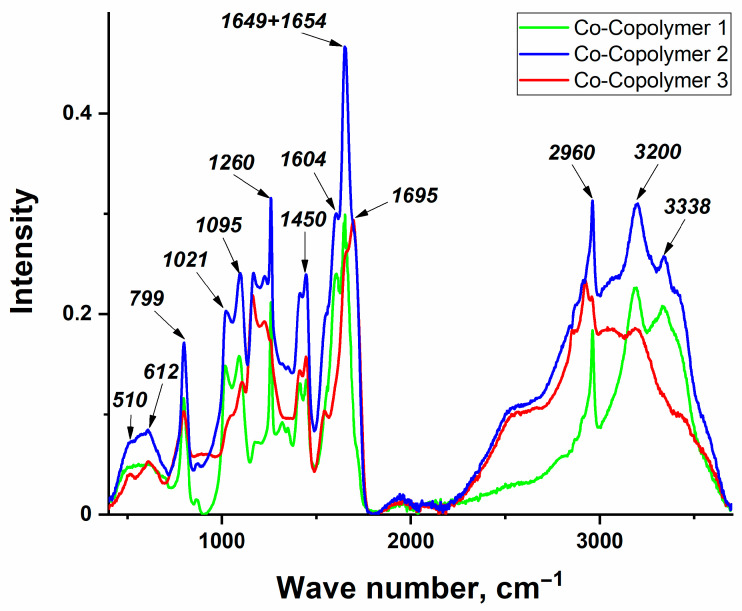
IR spectra of copolymers with 1 wt.% CoAcr_2_PhTpy: Co-Copolymer 1—AAm/AAc 84.5/14.5; Co-Copolymer 2—AAm/AAc = 49.5/49.5; Co-Copolymer 3—AAm/AAc = 14.5/84.5.

**Figure 3 polymers-16-03127-f003:**
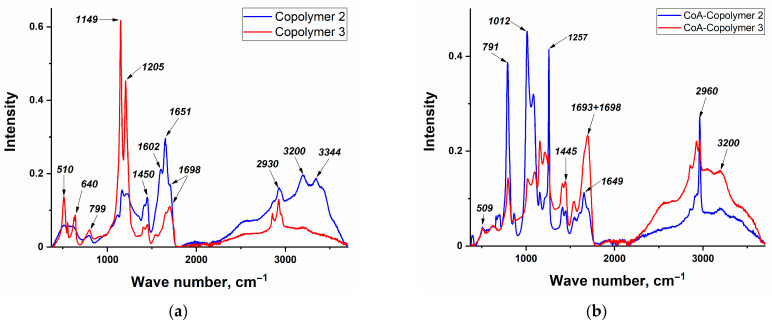
IR spectra of (**a**) «initial» copolymers Copolymer 2 (AAm/AAc = 50/50) and Copolymer 3 (AAm/AAc = 15/85) and (**b**) copolymers with CoAcr_2_: CoA-Copolymer 2—AAm/AAc = 49.5/49.5; CoA-Copolymer 3—AAm/AAc = 14.5/84.5.

**Figure 4 polymers-16-03127-f004:**
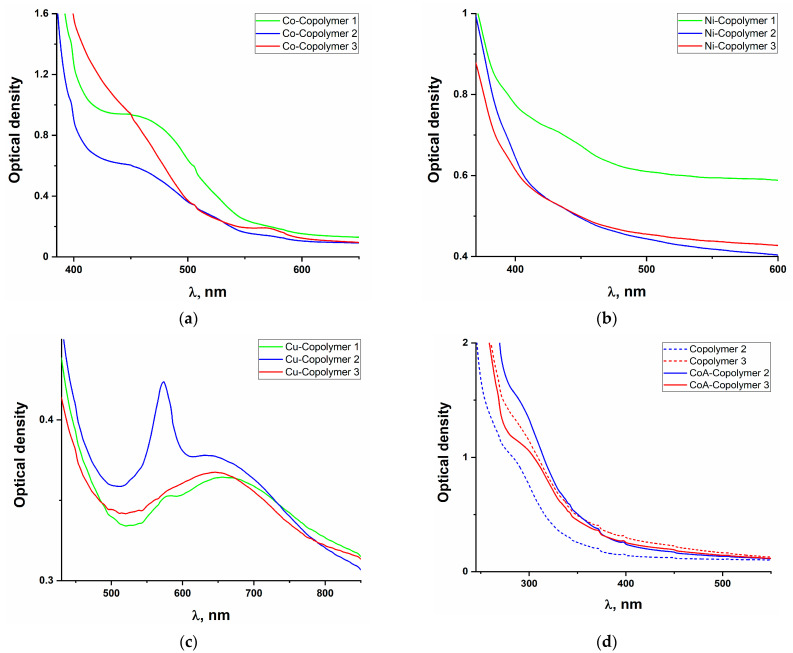
UV-visible spectra of (**a**)—Co-Copolymers, (**b**)—Ni-Copolymers, (**c**)—Cu-Copolymers, and (**d**)—model copolymers.

**Figure 5 polymers-16-03127-f005:**
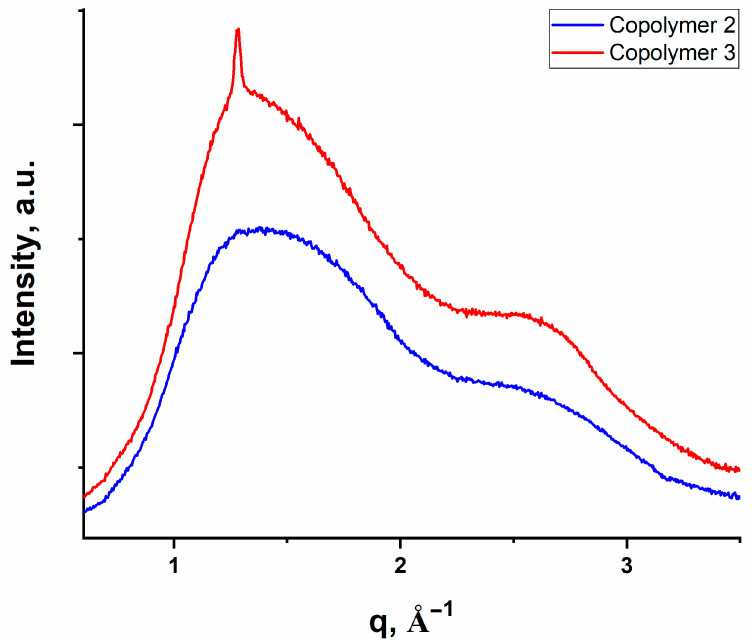
WAXS curves of model «initial» copolymers: Copolymer 2—AAm/AAc = 50/50; Copolymer 3—AAm/AAc = 15/85.

**Figure 6 polymers-16-03127-f006:**
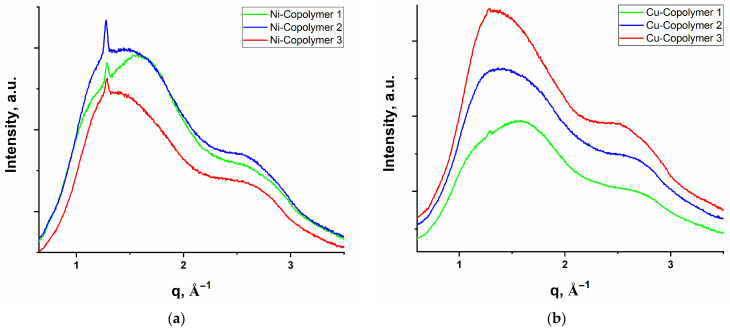
WAXS curves of copolymers (**a**) with 1 wt.% NiAcr_2_PhTpy and (**b**) with 1 wt.% CuAcr_2_PhTpy; monomer ratios for both systems: copolymer 1—AAm/AAc 84.5/14.5; copolymer 2—AAm/AAc = 49.5/49.5; copolymer 3—AAm/AAc = 14.5/84.5.

**Figure 7 polymers-16-03127-f007:**
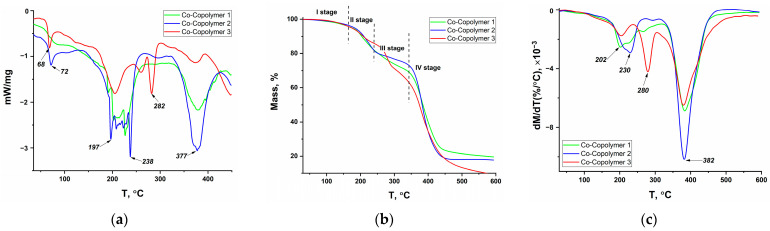
DSC (**a**), TGA (**b**) and DTG (**c**) curves of copolymers with CoAcr_2_PhTpy: Co-Copolymer 1—AAm/AAc 84.5/14.5; Co-Copolymer 2—AAm/AAc = 49.5/49.5; Co-Copolymer 3—AAm/AAc = 14.5/84.5.

**Figure 8 polymers-16-03127-f008:**
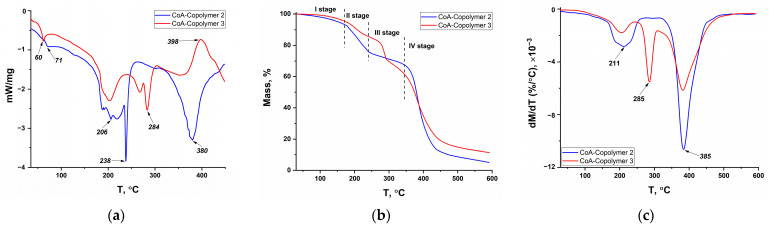
DSC (**a**), TGA (**b**) and DTG (**c**) curves of model copolymers with CoAcr_2_: CoA—Copolymer 2—AAm/AAc = 49.5/49.5; CoA-Copolymer 3—AAm/AAc = 14.5/84.5.

**Figure 9 polymers-16-03127-f009:**
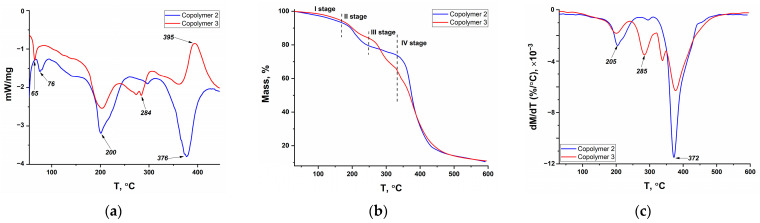
DSC (**a**), TGA (**b**) and DTG (**c**) curves of model copolymers without complexes: Copolymer 2—AAm/AAc = 50/50; Copolymer 3—AAm/AAc = 15/85.

**Figure 10 polymers-16-03127-f010:**
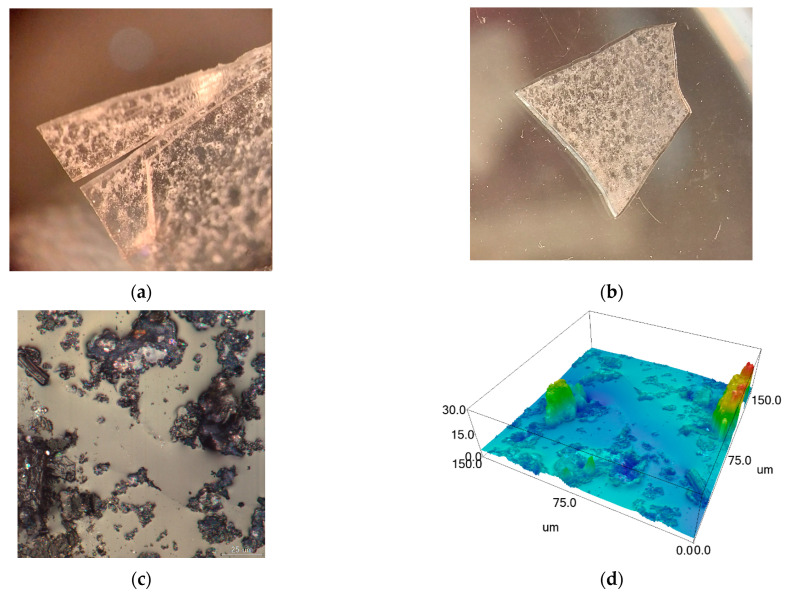
Autonomous intrinsic healing of Cu-Copolymer 2—optical microscopy images (**a**) before and (**b**) after healing; (**c**) laser scanning microscope image of the fracture area after healing and (**d**) con-focal image of its surface.

**Figure 11 polymers-16-03127-f011:**
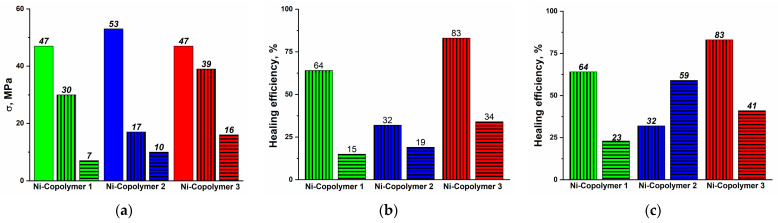
(**a**) Tensile strength of nickel complex copolymers (Ni-Copolymers), (**b**) healing efficiency relative to the first sample, and (**c**) relative efficiency after each healing. No bar represents the original sample, vertical bar represents the first sample healing, and horizontal bar represents the second sample healing.

**Table 1 polymers-16-03127-t001:** Composition of precursor solutions with different phenylterpyridine metal complexes.

Sample	Initial Weight, g					Ratios (%/%/%)
	AAm	AAc	H_2_O	KPS	MAcr_2_PhTpy	AAm/AAc/MAcr_2_PhTpy
Co-Copolymer 1	5.1	0.9	24	0.06	0.06	84.5/14.5/1
Co-Copolymer 2	3	3	24	0.06	0.06	49.5/49.5/1
Co-Copolymer 3	0.9	5.1	24	0.06	0.06	14.5/84.5/1
Ni-Copolymer 1	5.1	0.9	24	0.06	0.06	84.5/14.5/1
Ni-Copolymer 2	3	3	24	0.06	0.06	49.5/49.5/1
Ni-Copolymer 3	0.9	5.1	24	0.06	0.06	14.5/84.5/1
Cu-Copolymer 1	5.1	0.9	24	0.06	0.06	84.5/14.5/1
Cu-Copolymer 2	3	3	24	0.06	0.06	49.5/49.5/1
Cu-Copolymer 3	0.9	5.1	24	0.06	0.06	14.5/84.5/1

**Table 2 polymers-16-03127-t002:** Composition of precursor solutions of model systems.

Sample	Initial Weight, g					Ratios (%/%/%)
	AAm	AAc	H_2_O	KPS	CoAcr_2_	AAm/AAc/CoAcr_2_
Copolymer 2 [32]	10	10	20	0.2	-	50/50/0
Copolymer 3 [32]	3	17	20	0.2	-	14/85/0
CoA-Copolymer 2	3	3	24	0.06	0.024г	49.8/49.8/0.4
CoA-Copolymer 3	0.9	5.1	24	0.06	0.024г	14.9/84.7/0.4

**Table 3 polymers-16-03127-t003:** Results of elemental analysis and gel permeation chromatography of the obtained copolymers.

Sample	Content of Elements (%)				Ratios (%/%/%) AAm/AAc/MCM/Water		GPC Results		
	C	H	N	M	Copolymers	Copolymers **	M_w_	M_n_	PDI
CoA-Copolymer 2	42.9	6.9	8.2	0.024	41.0/43.3/0.1/15.5	48.6/51.3/0.1	-	-	-
CoA-Copolymer 3	44.7	6.5	2.3	0.026	11.4/75.2/0.1/13.3	13.1/86.8/0.1	-	-	-
Co-Copolymer 1	42.9	7.0	14.0	0.018	72.9/14.3/0.2/12.7	83.5/16.4/0.2	-	-	-
Co-Copolymer 2	44.6	6.7	8.8	0.021	44.5/44.1/0.2/11.2	50.1/49.7/0.2	18,000	14,300	1.3
Co-Copolymer 3	44.6	6.3	2.5	0.026	12.6/75.6/0.2/11.5	14.3/85.5/0.2	-	-	-
Ni-Copolymer 1	43.0	7.2	13.5	0.024	68.3/16.4/0.2/15.0	80.4/19.4/0.2	-	-	-
Ni-Copolymer 2	43.7	6.8	8.0	0.023	40.0/45.7/0.2/14.0	46.6/53.2/0.2	348,000	74,500	4.7
Ni-Copolymer 3	44.4	6.4	2.9	0.027	14.6/73.3/0.1/12.0	16.6/83.3/0.1	1,043,000	139,000	7.5
Cu-Copolymer 1	43.2	7.3	13.6	0.011	68.6/16.5/0.1/14.9	80.6/19.3/0.1	-	-	-
Cu-Copolymer 2	43.6	6.7	7.0	0.092	34.8/50.3/0.8/14.1	40.5/58.6/0.9	542,000	123,000	4.4
Cu-Copolymer 3	44.7	6.5	2.6	0.066	12.7/73.5/0.5/13.3	14.6/84.8/0.6	713,000	152,000	4.7

** Composition of copolymers excluding water.

**Table 4 polymers-16-03127-t004:** Tensile test results and self-healing efficiency.

Sample	Maximum Strength of the Original Sample, MPa	Maximum Strength of the Healed Sample, MPa	Z_σ_, % *	Modulus of Elasticity of the Original Sample, GPa	Modulus of Elasticity of the Healed Sample, GPa	Z_E_, %
CoA-Copolymer 2	82	8	** * 10 * **	5.3	3.1	58
CoA-Copolymer 3	100	19	** * 19 * **	5.9	5.2	88
Co-Copolymer 1	120	27	**23**	3.8	4.8	-
Co-Copolymer 2	87	14	** * 16 * **	5.5	2.6	47
Co-Copolymer 3	95	38	** * 40 * **	5.6	6.9	-
Ni-Copolymer 1	47	30	**64**	3.2	6.5	-
Ni-Copolymer 2	53	17	**32**	3.1	2.2	71
Ni-Copolymer 3	47	39	**83**	2.3	6.2	-
Cu-Copolymer 1	116	30	**26**	16.8	5.3	32
Cu-Copolymer 2	123	30	**24**	14.1	4.5	32
Cu-Copolymer 3	115	34	**30**	16.4	6.2	38

* Bold—main results to assess healing efficiency; italics and underline—comparative results to assess the effect of metal ion and phenylterpyridine ligand inclusion.

## Data Availability

The original contributions presented in the study are included in the article/Appendix A, further inquiries can be directed to the corresponding author.

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
