# Peer review of "High-Strength, Self-Healing Copolymers of Acrylamide and Acrylic Acid with Co(II), Ni(II), and Cu(II) Complexes of 4′-Phenyl-2,2′:6′,2″-terpyridine: Preparation, Structure, Properties, and Autonomous and pH-Triggered Healing"

_polymers, 2024, doi:10.3390/polym16223127_

Round 1

Reviewer 1 Report

Comments and Suggestions for Authors

This paper describes self-healing copolymers of acrylamide, acrylic acid and acrylic complexes of phenyl-terpyridine [Co(II), Ni(II), Cu(II)]. These results are extension of authors previous work about similar type of copolymers with Co(II), Zn(II), Cu(II) (references 31, 32). However, newly synthesized copolymers with nickel complex (Ni-Copolymers) showed the best self-healing ability among metallopolymer series. Therefore, this paper might be suitable for publication in Polymer after addressing some comments described below.

The authors should pay attention to the purity of the copolymers. Some copolymers such as Ni -copolymer contain potassium persulfate used as initiator as confirmed by WAXS, but this is almost negligible in the case of Cu-copolymer. The influence of metal effect on the self-healing property is the main topic of this paper. The impurity of metal may have influence on mechanical property. If possible, potassium-free copolymers should be used for comparison.

Author Response

Dear Reviewer 1,

We are grateful for time and effort that you paid for reviewing our article. All the comments have helped us to greatly improve the manuscript. Here is a point-by-point response to comments and concerns.

Comment 1: The authors should pay attention to the purity of the copolymers. Some copolymers such as Ni -copolymer contain potassium persulfate used as initiator as confirmed by WAXS, but this is almost negligible in the case of Cu-copolymer. The influence of metal effect on the self-healing property is the main topic of this paper. The impurity of metal may have influence on mechanical property. If possible, potassium-free copolymers should be used for comparison.

Response 1: Thank you very much for the important observation. It is indeed the case that a peak is observed on the WAXS curves for nickel copolymers, which most likely refers to the potassium persulfate signal. It is unclear why only a subset of the samples exhibit this signal. Nevertheless, it seems reasonable to assume that potassium persulfate will have no effect on the mechanical properties or healing efficiency.

The primary factor influencing the mechanical properties of these systems is the incorporation of a divalent metal ion into the polymer chain, due to the presence of two acrylate groups. Additionally, the potential for further coordination interactions, characteristic of transition metals (Co(II), Ni(II), Cu(II)), may also play a role. It is not feasible for potassium ions to function as a bonding centre between the links of two chains.

As noted in the paper, the main contribution to self-healing is attributed to the processes of coordination (transition metal ion with amide and carboxyl groups) and dissociation (metal ions in the main polymer chain with side carboxyl groups, see scheme 1). It is not possible for potassium ions (I) to form a bond with two carboxyl groups upon dissociation, as is the case with divalent cobalt, nickel or copper ions. Furthermore, they are unable to engage in complexation reactions.

A comparison with copolymers devoid of potassium ions would undoubtedly prove beneficial for a more comprehensive analysis and substantiation of the aforementioned assumptions. Nevertheless, the process of obtaining the copolymer and its film, as well as studying its self-healing properties and performing WAXS analysis, is a time-consuming endeavour. A study of this nature will be conducted at a future date.

Scheme 1. Probable mechanism of healing due to bonds breaking and formation by dissociation.

Reviewer 2 Report

Comments and Suggestions for Authors

This work investigates metallopolymers that demonstrate self-healing properties while maintaining good mechanical properties. Overall, the paper is well-written and experiments have a logical flow. However, there is a lack of mechanistic studies or even discussion on how self-healing occurs. Specific comments below:

Method section – Film formation is not mentioned but only referenced to a previous paper. Details should be included here as well. This appears to be a cross-linked polymer based on the difunctional metallo-monomer. Does this mean that the film is cast before the reaction is complete? Or is there a specific mold the reaction is performed on?

Line 260 – “GPC was performed on ash part”…The word “ash” was mentioned several times in the manuscript, please describe in detail what does this refer to and method of preparation.

Line 286-287 – “Co-Copolymer 2 and take the minimum values among all obtained. This is probably due to the highest reactivity of the cobalt complex during copolymerization” Can the authors explain in the manuscript why does the cobalt complex poses the highest reactivity?

The self-healing properties of the metallopolymers are impressive especially the Ni-based polymer, however, there is a lack of mechanistic study here. I understanding that this might be an ongoing project but the authors need to at least show a logical scheme of how the coordination complex contributes to self-healing.

Furthermore, how does HCl accelerate the healing process? It is stated that it protonates the sides groups and increases hydrogen bonding, but these already exist in the initial copolymer. Why does adding HCl in the metallopolymer improve self-healing but not in the initial copolymer?

Author Response

Dear Reviewer 2,

We are grateful for time and effort that you paid for reviewing our article. All the comments have helped us to greatly improve the manuscript. Here is a point-by-point response to comments and concerns.

Comment 1: Method section – Film formation is not mentioned but only referenced to a previous paper. Details should be included here as well. This appears to be a cross-linked polymer based on the difunctional metallo-monomer. Does this mean that the film is cast before the reaction is complete? Or is there a specific mold the reaction is performed on?

Response 1: Thank you for your question. The process of obtaining the films was conducted in accordance with the methodology outlined in the paper (lines 181-184), namely: «The films were formed from an aqueous polymer solution in special open glass molds by air drying at room temperature. The drying process was carried out until the film mass reached a constant value. Once the film had completely detached from the mold, it was inverted to ensure maximum drying on both sides».

Thus, it was the polymer solution that was poured into special glass moulds. Perhaps there has been a misunderstanding, so it might be helpful to clarify a couple of points.

It is generally accepted that copolymerization with a bifunctional monomer results in the cross-linking of polymer chains, which precludes the possibility of dissolving the cross-linked polymer to create films. For these systems, however, it is possible. This may be related to the dissociation of the metal ion (Scheme 1), which serves as the 'cross-linking centre' (lines 594-601) (the same phenomenon probably also plays an important role in healing processes), and to the introduction of additional free volume due to the phenylterpyridine ligand (lines 279-282). These phenomena make it possible to obtain films.

Scheme 1. Probable mechanism of healing due to bonds breaking and formation by dissociation.

Comment 2: Line 260 – “GPC was performed on ash part”. The word “ash” was mentioned several times in the manuscript, please describe in detail what does this refer to and method of preparation.

Response 2: The ash part is the soluble part. The polymer was dissolved at room temperature for 24 h, followed by a further 6 h at 55°C (we have previously selected a sample preparation technique for GPC analysis). The polymer solution (ash part) was then carefully separated from the undissolved part of the polymer (precipitate/gel part) using a syringe. The polymer solution was further filtered, after which the filtrate was used for chromatography.

Added to the experimental part (lines 203-207).

Comment 3: Line 286-287 – “Co-Copolymer 2 and take the minimum values among all obtained. This is probably due to the highest reactivity of the cobalt complex during copolymerization” Can the authors explain in the manuscript why does the cobalt complex poses the highest reactivity?

Response 3: As previously demonstrated [1) Dzhardimalieva, G.I., Pomogailo, A.D. (1990). Variability of Mixed-Unit Chains in Metal-Containing Polymers. In: Pittman, C.U., Carraher, C.E., Zeldin, M., Sheats, J.E., Culbertson, B.M. (eds) Metal-Containing Polymeric Materials. Springer, Boston, MA. https://doi.org/10.1007/978-1-4613-0669-6_4; 2)  Dzhardimalieva, G.I.; Pomogailo, A.D. Macromolecular Metal Carboxylates. Russ. Chem. Rev. 2008, 77, 259–301, https://doi.org/10.1070/RC2008v077n03ABEH003682], the distinctive characteristics of metal acrylates are intimately associated with the nature of the d-element. The study of liquid-phase radical polymerization has demonstrated that the rate of polymerization of metal acrylates is slower than that of the homopolymerization of the «metal-free» analogue, acrylic acid. Furthermore, this rate decreases in series: AAc>Co2+> Ni2+>Fe3+>Cu2+.

The change in the rate of polymerization of metal acrylates in the above series is explained by the electronic effect of the metal (depending on the value of its electronegativity) on the reactivity of the unsaturated bond, as well as by the change in the degree of conjugation of the unsaturated bond to the carboxylate ion depending on the nature of the metal. A similar effect is observed in the hydrogenation reactions of metal acrylates [Pomogailo, A.D.; Dzhardimalieva, G.I. Problems of Unit Variability in Metal-Containing Polymers. Russ Chem Bull 1998, 47, 2319–2337, https://doi.org/10.1007/BF02641530], whereby it is demonstrated that the hydrogenation rate declines in accordance with the increasing electronegativity of the metal. The data obtained also correlate with data from another study [Czerniawski, T.; Wojtczak, Z. Determination of the Initiation Rate in the Polymerization of Zinc, Cobalt and Nickel Acrylates. European Polymer Journal 1996, 32, 1035–1036, https://doi.org/10.1016/0014-3057(96)00031-6], which demonstrated that the activation energy for the polymerization of cobalt acrylate is lower than that observed for other acrylates, such as nickel acrylate.

The explanation added to the paper (lines 300-305).

Comment 4-5: The self-healing properties of the metallopolymers are impressive especially the Ni-based polymer, however, there is a lack of mechanistic study here. I understanding that this might be an ongoing project but the authors need to at least show a logical scheme of how the coordination complex contributes to self-healing.

Furthermore, how does HCl accelerate the healing process? It is stated that it protonates the sides groups and increases hydrogen bonding, but these already exist in the initial copolymer. Why does adding HCl in the metallopolymer improve self-healing but not in the initial copolymer?

Response 4-5: The ability for autonomous intrinsic healing, in addition to the ability to generate homogeneous films of the resulting metallopolymer, is presumably attributable to the dissociation processes of metal acrylate links within the polymer chain (Scheme 1). Dissociation can result in the reversible formation of new bonds between functional groups of different polymer chains (reversible binding via carboxyl groups is indicated in the scheme), as well as an increase in the kinetic flexibility of the polymer chains as a whole. In this regard, the incorporation of metal ions into the polymer chain results in the phenomenon of autonomous intrinsic healing, which is in contrast to the model films of AAm:AAc copolymers.

The presence of phenylterpyridine ligand can increase the efficiency of the healing process due to several factors. The presence of a strong coordinating ligand can lead to delocalization of electron density at the metal ion, thus facilitating its dissociation processes. In addition, the sterically hindered ligand increases the free volume of the polymer, which may facilitate solvent diffusion during healing. Furthermore, the incorporation of such a substituent frequently results in the formation of weak, non-covalent, reversible interactions between the π-bonds of aromatic rings in the polymer structure (π-π-stacking). These factors collectively enhance the efficacy of the healing process. A more detailed study of the healing mechanism of these systems is planned for the near future. This will be carried out using theoretical calculations and dynamic modelling.

In regard to pH-triggered healing, it typically results in the protonation of polymer functional groups, thereby introducing supplementary weak electrostatic interactions into the existing bonding system, as a result of which polymer chains can bind via carboxyl and amide groups. Therefore, pH-triggered healing was also observed in model copolymers (AAm:AAc), as had been demonstrated in previous research, albeit with low efficiency values (up to 10%). In metallopolymers, pH-triggered can also facilitate the release of metal ions from the polymer chain, effectively accelerating the processes shown in Scheme 1, ultimately leading to higher healing rates, which translates into a multiple increase in healing efficiency.

It seems probable that the release of metal ions upon the initiation of HCl plays a more significant role in the case of metallopolymers. However, this acceleration mechanism requires further investigation, including theoretical calculations, which we intend to undertake in the future.
